# Terahertz Combined with Metamaterial Microfluidic Chip for Troponin Antigen Detection

**DOI:** 10.3390/mi13122257

**Published:** 2022-12-19

**Authors:** Yen-Shuo Lin, Shih-Ting Huang, Shen-Fu Hsu, Kai-Yuan Tang, Ta-Jen Yen, Da-Jeng Yao

**Affiliations:** 1Institute of NanoEngineering and MicroSystems, National Tsing Hua University, Hsinchu 30013, Taiwan; 2Department of Power Mechanical Engineering, National Tsing Hua University, Hsinchu 30013, Taiwan; 3ACE BIOTEK Co., Ltd., Hsinchu 30261, Taiwan; 4Department of Material Science Engineering, National Tsing Hua University, Hsinchu 30013, Taiwan

**Keywords:** terahertz, metamaterials, microfluidics, troponin, antigen

## Abstract

In this paper, we use terahertz combined with metamaterial technology as a powerful tool to identify analytes at different concentrations. Combined with the microfluidic chip, the experimental measurement can be performed with a small amount of analyte. In detecting the troponin antigen, surface modification is carried out by biochemical binding. Through the observation of fluorescent antibodies, the average number of fluorescent dots per unit of cruciform metamaterial is 25.60, and then, by adjusting the binding temperature and soaking time, the average number of fluorescent dots per unit of cruciform metamaterial can be increased to 181.02. Through the observation of fluorescent antibodies, it is confirmed that the antibodies can be successfully stabilized on the metamaterial and then bound to the target antigen. The minimum detectable concentration is between 0.05~0.1 μg/100 μL, and the concentration and ΔY show a positive correlation of R^2^ = 0.9909.

## 1. Introduction

The terahertz (THz) gap generally refers to the frequency band in the range of 0.1 to 10.0 THz, located between the microwave and infrared regions of the electromagnetic spectrum [1]. In addition, terahertz radiation can penetrate without destroying most non-polar materials, similar to the properties of microwaves, and has applications in safety testing due to its ability to distinguish objects [2]. THz radiation can be used to identify various biochemical structures and weak interactions, including hydrogen bonds and van der Waals forces in the terahertz range, and low-frequency vibrations and rotations of biomolecules can be probed with terahertz spectroscopy [3]. In addition, the specific fingerprint frequencies of most molecules, including proteins, DNA, and viruses, are located in the THz region, and non-destructive sensing of lower energy terahertz radiation can be acquired [4]. Due to its unique properties, THz technology is expected to be used to examine biological samples for label-free, non-contact, and non-destructive testing [5,6].

Metamaterials are composed of a series of subwavelength structures and belong to artificial engineering materials. The properties of metamaterials can be determined by altering the material properties of the surrounding medium or the resonator geometry [7]. When metamaterials interact with electromagnetic waves with wavelengths that far exceed the distance between metamaterials components, metamaterials can become effective electromagnetic scattering media [8]. Once the metamaterial interacts with electromagnetic waves, a frequency resonance with a high-quality factor is formed, and the resonance causes a strong absorption effect. In the frequency spectrum, the signal is sometimes observed to drop sharply and then rise, like a pointer in the frequency domain. When the dielectric environment changes, the signal can be obtained by the resonant shift value [9]. By patterning metallic nanostructures in arrays, ultrasensitive bio-detection, drug delivery, cancer monitoring, and biomedical imaging can be achieved [10].

With the development of microfluidic chips, because of the mature technology of micro-electromechanical systems (MEMS), the size of the lithography process has been reduced to micrometers or even nanometers. Through the microfluidic chip, the amount of reagents used can be reduced, and various experimental processes in traditional biomedical laboratories can be combined, such as solution mixing, separation, and detection, to make microchips to achieve lab-on-chips (LOC) [11].

For the detection of biological samples, such as antigens, antibodies, fungal cells, and biological proteins, both the detection system using polymerase chain reaction (PCR) and the fluorescence-based microbial detection system are generally considered to be effective methods for visual tracking and monitoring of biomolecular compounds [11]. However, these methods have the disadvantage of being time-consuming, expensive, and requiring label-intensive detection. For acute myocardial infarction (AMI), detecting such a disease depends on blood analysis [12]. Traditional methods rely on relatively high concentrations of enzyme proteins so they can be detected. If such high concentrations are acquired, it takes at least 6 h from the onset of symptoms to the release of troponin I into the blood sample. However, with the sensing capabilities of terahertz systems, combined with metamaterials and microfluidic systems, enabling rapid, label-free, and non-destructive detection will become an emerging biomedical detection method [13].

## 2. Microfluidic System, Metamaterials, and Experimental Setup

### 2.1. Microfluidic System and Metamaterial

In the future application of the chip in the biomedical field, it is important to select the appropriate material for the microfluidic chip, which needs to be biocompatible and can be fabricated into a microfluidic chip structure. The polymer material (PP) is selected for production, which is not only compatible with biological samples but also has a greater rigidity than the common microfluidic material (PDMS) in the production of chips, which can avoid the superposition of the upper and lower layer substrates. 

The design of the microfluidic system is shown in Figure 1a. The height of the microfluidic channel is designed to be 10 µm, where the channel material is cut out in the middle detection area, and the thickness of the material is reduced to 0.5 mm to reduce the influence of the microfluidic chip on THz absorption. The chip is shown in Figure 1b; the upper and lower layers of the microfluidic are fabricated by injection molding and then bonded by ultrasonic melting to produce a complete microfluidic system.

When specific metamaterials interact with THz radiation, metamaterial resonance occurs in the frequency spectrum. When THz is perpendicularly incident on a metamaterial with a four-axis pattern, its quadrupole resonance mode will generate a quadrupole resonance, which will cause reflection cancelation, resulting in high absorption and obtaining a high-quality factor and high sensitivity [14,15]. 

As shown in Figure 1c, on the four-axis pattern design, a cruciform metamaterial is designed with a length of 220 μm, a width of 75 μm, and an angle of 90 degrees. Then, the cruciform metamaterial unit is deposited at the height of 600 nm by silver metal sputtering and is arranged into a metamaterial array with a period length (periodicity) of 350 μm. Finally, the metamaterial array is designed in the detection area of the chip to resonate with THz. As shown in Figure 1d, the experimental resonance position of the chip containing the metamaterial array is at 485.63 GHz, and the spectrum presents a resonance dip, which means a larger figure of merit (FOM) and quality factor [16]. This unique resonance dip is used as an indicator to conduct quantitative and qualitative analyses by observing the *x*- and *y*-axis resonance shift and variation in the resonant dip.

### 2.2. Experimental Setup

The experimental setup is shown in Figure 2a, where the THz radiation penetrates the chip vertically, and when the analyte exists on the surface of the metamaterial, the metamaterial environment has a dielectric constant difference, and the signal received by the detector changes. Finally, through the frequency domain signal, observe the *x*- and *y*-axis resonance shift and variation in the resonant dip.

The THz instruments are the TeraPulse system (TeraPulse 4000) and the VDI system. As shown in Figure 2b, the TeraPulse system is THz time-domain spectroscopy (THz-TDS), and the generation of the terahertz-pulsed radiation is based on a photoconductive switch. Its operating regime is from 100 GHz to 4000 GHz, and its minimum resolution is 7.09 GHz. The TeraPulse system is a closed measuring instrument. After the chip is fixed on the platform in the chamber, the chamber will be sealed. The chamber will extract vapor and inject dry air to reduce the effect of vapor on the absorption of terahertz waves. As shown in Figure 2c, the wave source of the VDI system is the continuous wave generated by the RF electron beam, and the detector receives the signal in the frequency domain. It has a narrow regime of 360 GHz to 500 GHz and a higher resolution of 0.36 GHz.

## 3. Troponin Experiment

### 3.1. Silver Surface Modification Method

Troponin I is an antigen associated with acute myocardial infarction (AMI) disease. As a property of a wide range of proteins, it is generally caught with a specific antibody. In order to immobilize the antigen on the metal surface, surface modification is required so that the antigen can bind to the metal through chemical bonds.

In order to modify the silver surface, a chemical structure is formed on the surface, as shown in Figure 3, using 11-mercaptoundecanoic acid (11-MUA) soaking to avoid harmful oxidation of the silver layer. After soaking, excess unbound 11-mercaptoundecanoic acid (11-MUA) was washed away with dimethyl sulfoxide (DMSO), and the 11-MUA film was attached to the silver film. Before binding the antibody, the metal must first be modified to bind the antibody to the surface, thus soak with 1-ethyl-3-(3-dimethylaminopropyl)carbodiimide/N-hydroxysuccinimide (EDC/NHS) solution to obtain a surface modification. EDC/NHS was added to make the surface reactive so that the antibody could bind with the surface. After soaking, the excess EDC/NHS was washed away with phosphate-buffered saline (PBS buffer). After the modification, silver metal can then bind to the antibody. Finally, the antigen was added to generate a specific immune reaction [17].

### 3.2. Surface Modification Process of Fluorescent Antibody Inspection

In order to confirm the effectiveness of the modification process, the steps of 11-MUA and EDC/NHS are the keys to the whole experiment. The fluorescent antibody is used to react with the modified chip. After the reaction is completed, the number of fluorescent dots is observed under a fluorescent microscope. The fluorescent antibody is FITC conjugate, the corresponding excitation light wavelength is 488 nm, and the green light wave is visible to the naked eye.

The fluorescent antibody experimental procedure included the following steps: (1) Soak the chip in 10 mM MUA in DMSO for 24 h; (2) the excess MUA is washed away with DMSO solution; (3) EDC/NHS is prepared into a 0.1 M:0.05 M DMSO solution. After mixing with a vortex, soak the chip in the solution for 1 h; (4) the excess EDC/NHS is washed away with PBS solution; (5) take 5 μL of fluorescent antibody solution, drop it on the surface of the silver on the chip, and react for 1 h at room temperature in a lithography laboratory to prevent the degradation of the fluorescent dot; (6) the excess fluorescent solution is washed away with PBS solution; (7) and after completing the above steps, observe the number of fluorescent dots under a fluorescence microscope.

The experimental results of the fluorescent antibody are shown in Figure 4a. In order to calculate the results, the Image J software was used to analyze the experimental results and calculate the upper, left, middle, right, and lower areas of the metamaterial array (areas A, B, C, D, and E). Three metamaterials were taken in each region to calculate the number of fluorescent dots.

Figure 4b,c are the fluorescence results of the comparison with and without 11-MUA. In Figure 4b, the silver cruciform area is framed by the yellow line. In the non-cruciform pattern area, since the PP substrate does not participate in the surface modification reaction, the fluorescent antibody cannot be bound to it, and no fluorescent dots are observed. In the cruciform pattern area, because the silver metamaterial participates in the surface modification reaction, the fluorescent antibody can be bound to it, and it can be observed that there are bright fluorescent dots on the silver surface. The average number of fluorescent dots in the cruciform in this experiment is 25.60. Figure 4c is the result without using 11-MUA, and the average number of fluorescent dots in the cruciform in this experiment is only 7.33, which proves that the use of 11-MUA can effectively improve the effect of surface modification.

### 3.3. Experiment Parameter Optimization and Adjustment

In the previous experiment, the average number of cruciform fluorescent dots was 25.60. In order to make the subsequent binding antigen more significant, the experimental parameters of the fluorescent antibody were optimized and adjusted. The results of the antibody parameter optimization experiment are shown in Figure 5a. Chip A is placed at a room temperature of 23 degrees Celsius, and the soaking time is 1 h (original parameters). Chip B is placed at a refrigerated temperature of 4 degrees Celsius, and the soaking time is 1 h (reduced antibody soaking temperature). Chip C is placed at a room temperature of 23 degrees Celsius, and the soaking time is 24 h (extended soaking time). Chip D is placed at a refrigerated temperature of 4 degrees Celsius, and the soaking time is 24 h (reduced soaking temperature and extended soaking time—change the antibody soaking temperature and extend the antibody soaking time for the experimental observation). The calculation results of the number of fluorescent dots are shown in Figure 5b; the average number of fluorescent dots for chips A, B, C, and D are 25.60, 93.60, 103.27, and 183.13. The reason for reducing the antibody soaking temperature is that the fluorescent antibody has the best reaction temperature at 4 degrees Celsius, and the antibody soaking time is extended for the experiments to improve the effect of antibody binding.

In terms of the EDC/NHS parameters, the related pieces of literature on surface modification were surveyed. Three works of literature used EDC/NHS for surface modification, and their concentrations were 100 mM EDC + 50 mM NHS [17], 10 mM EDC + 10 mM NHS [18], and 8 mM EDC + 10 mM NHS [19], respectively. As shown in Figure 6, 100 mM EDC + 50 mM NHS uses chips E and F; 10 mM EDC + 10 mM NHS uses chips G and H; 8 mM EDC + 10 mM NHS uses chips I and J for the experiments. The average fluorescent dots of chips E and F are 181.02 and 191.77; the average fluorescent dots of chips G and H are 176.20 and 172.00; the average fluorescent dots of chips I and J are 159.34 and 163.81. It was observed that the EDC/NHS concentration at 100 mM EDC + 50 mM NHS had the best binding effect during the surface modification.

### 3.4. Experiment Results of Terahertz Detection of Troponin Antigen

After the fluorescent antibody experiment and the optimization of the experimental parameters, it was confirmed that the antibody could indeed bind to the chip. Therefore, all modification procedures are carried out in the chip, and then the antibody is added to bind the target antigen in specificity. 

The troponin antigen experimental procedure included the following steps: (1) 10 mM MUA in DMSO was injected into the chip for 24 h; (2) the excess MUA was washed away with the DMSO solution; (3) EDC/NHS was prepared into a 0.1 M:0.05 M DMSO solution. After mixing with a vortex, EDC/NHS was injected into the chip for 1 h; (4) the excess EDC/NHS was washed away with the PBS solution; (5) 5.0 μg/100 μL of troponin antibody was injected into the chip, stored at 4 degrees Celsius and soaked for 24 h; (6) troponin antibody was washed away with the PBS solution; (7) six concentrations of troponin antigen were injected into the chip, stored at 4 degrees Celsius and soaked for 24 h; (8) troponin antigen was washed away with the PBS solution; (9) and the measurement was performed after evaporating the water in the chip. In the troponin antigen concentration section, six different concentrations of troponin antigen (2.0 μg/100 μL, 1.5 μg/100 μL, 1.0 μg/100 μL, 0.5 μg/100 μL, 0.1 μg/100 μL, and 0.05 μg/100 μL) were injected into the chip. Three chips were used for each of the troponin antigens at different concentrations for repeatable experiments.

The measurement results from the TeraPulse instrument are shown in Figure 7. Each concentration figure has four resonance curves. The resonance curves from bottom to top are the initial position, after surface modification, after adding antibodies and adding different concentrations of antigen resonance curves. Under the TeraPulse measurement with a resolution of 7.09 GHz, the *x*-axis resonance positions are all located at 485.63 GHz; thus, the next step is to identify the *y*-axis resonance position variation.

Table 1 shows the *y*-axis resonance position variation (ΔY) during the experiment. After the surface modification and the addition of the antibody, the ΔYs are between 0.0039~0.0040 a.u. and 0.0029~0.0030 a.u. After adding the antigens at concentrations of 2.0 μg/100 μL, 1.5 μg/100 μL, 1.0 μg/100 μL, 0.5 μg/100 μL, 0.1 μg/100 μL, and 0.05 μg/100 μL, the ΔYs were 0.0023 a.u., 0.0017 a.u., 0.0012 a.u., 0.0006 a.u., 0.0002 a.u., and 0.0000 a.u. As the antigen concentration decreased, the ΔY gradually decreased. Figure 8 shows the relationship between the antigen concentration and the ΔY measured by the TeraPulse. The minimum concentration that can be detected by the TeraPulse is between 0.05~0.1 μg/100 μL, and the antigen concentration and ΔY show a positive correlation of R^2^ = 0.9909.

Figure 9 is the measured resonance curves of six different concentrations of troponin antigens in the VDI System. The concentrations were also 2.0 μg/100 μL, 1.5 μg/100 μL, 1.0 μg/100 μL, 0.5 μg/100 μL, 0.1 μg/100 μL, and 0.05 μg/100 μL. The figure shows resonance curves for both the initial position and after adding different concentrations of antigen. From left to right, Table 2 displays the *x*- and *y*-axis resonance positions at the initial position, the *x*- and *y*-axis resonance positions after adding the antigen, the *x-* and *y*-axis resonance shift and variation, and the *x*- and *y*-axis average resonance shift and variation.

On the average shift of the *x*-axis (ΔX), the concentration of 0.00~0.5 μg/100 μL has a 0.72 GHz shift, 1.0~1.5 μg/100 μL has a 1.08 GHz shift, and 2.0 μg/100 μL has a 1.44 GHz shift. On the *y*-axis average variation (ΔY), the variation demonstrates 0.0026, 0.0026, 0.0026, 0.0036, 0.0047, 0.0054, and 0.0065 a.u. changes from low to high concentrations. Figure 10 shows the relationship between the concentration of antigen and ΔX, ΔY. The minimum concentration that can be detected by the chip in the VDI system is between 0.1~0.5 μg/100 μL, and the concentration and ΔY show a positive correlation of R^2^ = 0.9743.

## 4. Conclusions

Microfluidic chips can perform a sample analysis with a small amount of analyte, which is undoubtedly the future trend in the field of biomedicine. In addition, the detection method uses terahertz combined with metamaterial technology to check biological samples in order to achieve label-free, non-contact detection. Furthermore, through the biochemical binding method, the metamaterial is surface modified step by step, and the troponin antibody and antigen are bound. Through the adjustment of experimental parameters, the average number of fluorescent dots per unit of cruciform metamaterial can be increased to 181.02, and the minimum detectable concentration of the chip is 0.05~0.1 μg/100 μL. Terahertz’s microfluidic chip, combined with metamaterial, can be developed towards high-sensitivity biosensors in the future, which can be applied to the detection and analysis of different analytes.

## Figures and Tables

**Figure 1 micromachines-13-02257-f001:**
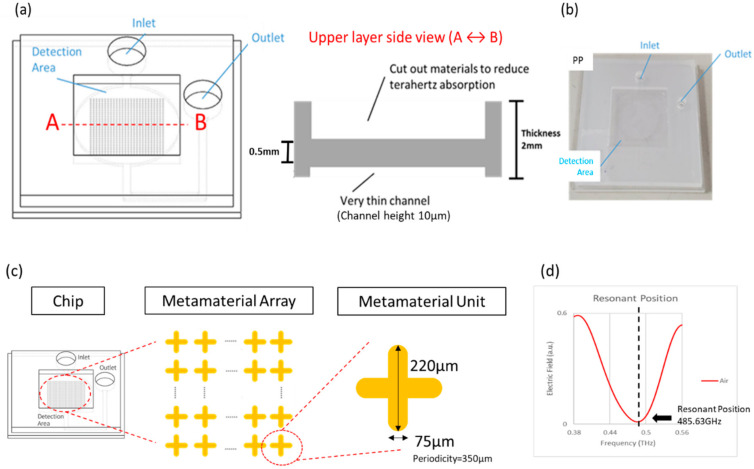
(**a**) The design of the microfluidic system; (**b**) complete microfluidic chip; (**c**) the design of the cruciform silver metamaterial; (**d**) experiment result shows a resonant dip at frequency 485.63 GHz.

**Figure 2 micromachines-13-02257-f002:**
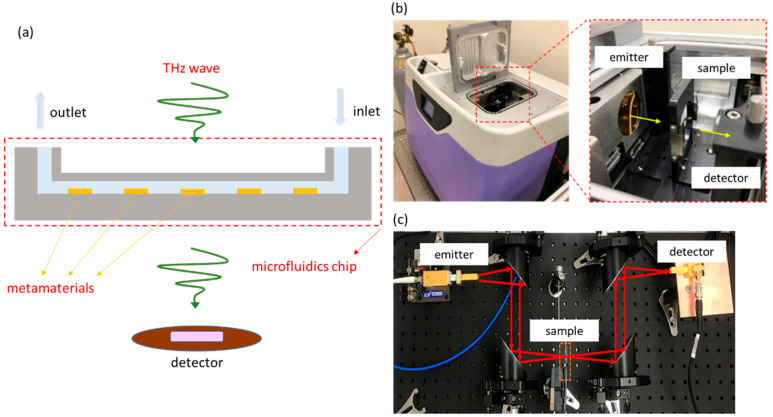
(**a**) Schematic diagram of experimental setup; (**b**) TeraPulse system structure diagram; (**c**) VDI system structure diagram.

**Figure 3 micromachines-13-02257-f003:**
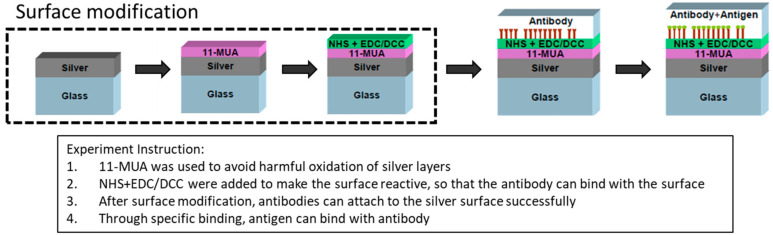
Flow chart of surface modification of silver film.

**Figure 4 micromachines-13-02257-f004:**
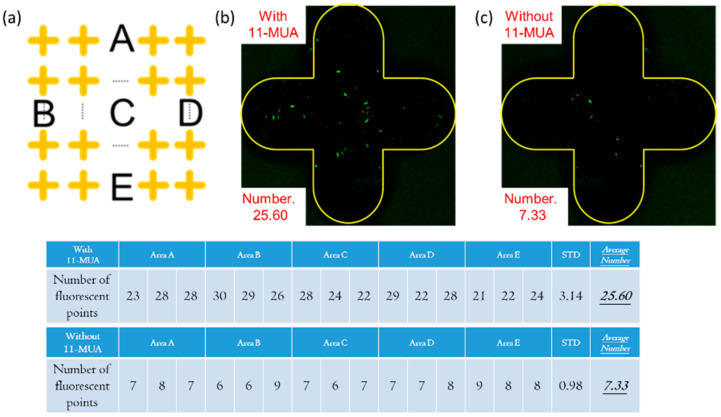
(**a**) Five regions on the top, left, middle, right, and bottom of the metamaterial array; (**b**,**c**) the average number of fluorescent dots results, with and without 11-MUA.

**Figure 5 micromachines-13-02257-f005:**
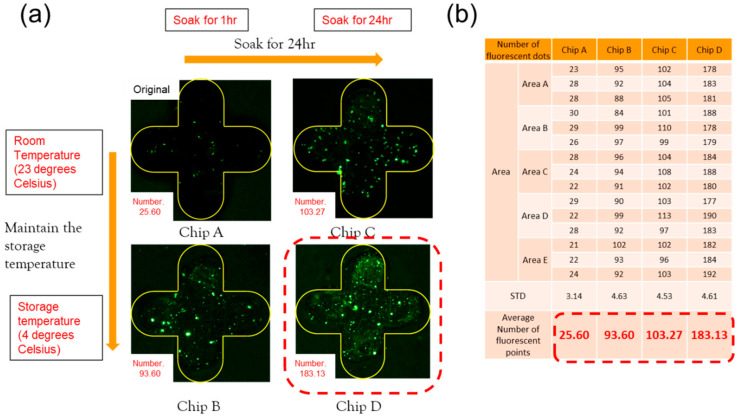
(**a**) Fluorescent dot picture; (**b**) number statistics of antibody parameter optimization experiment.

**Figure 6 micromachines-13-02257-f006:**
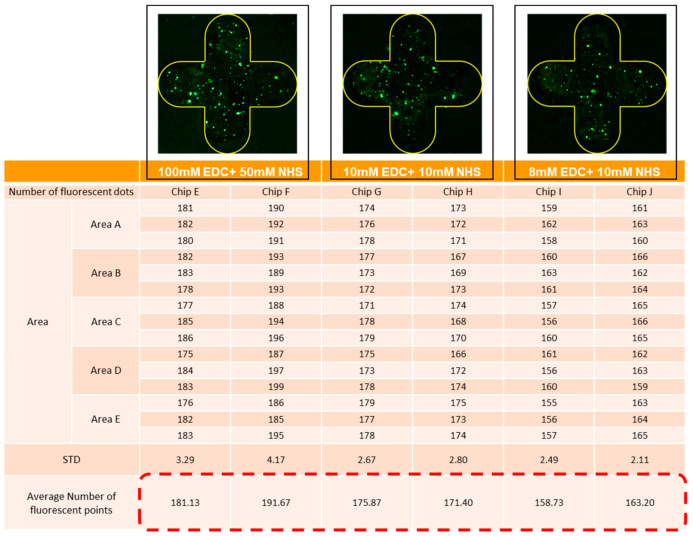
Experimental results and fluorescence quantity of three EDC/NHS parameters.

**Figure 7 micromachines-13-02257-f007:**
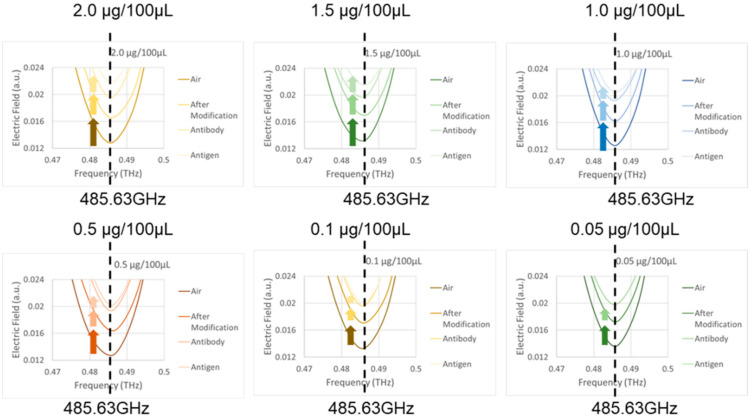
Resonance curve of troponin antigen experiment by TeraPulse.

**Figure 8 micromachines-13-02257-f008:**
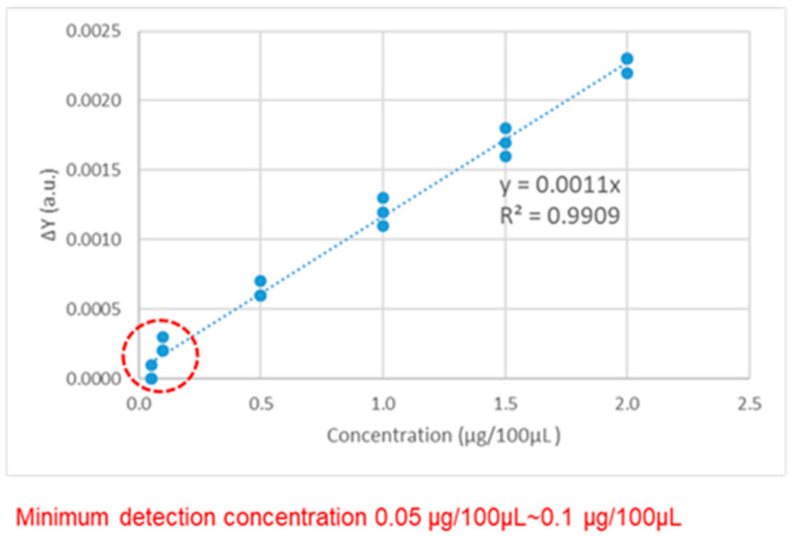
Relationship between the antigen concentration and ΔY by TeraPulse.

**Figure 9 micromachines-13-02257-f009:**
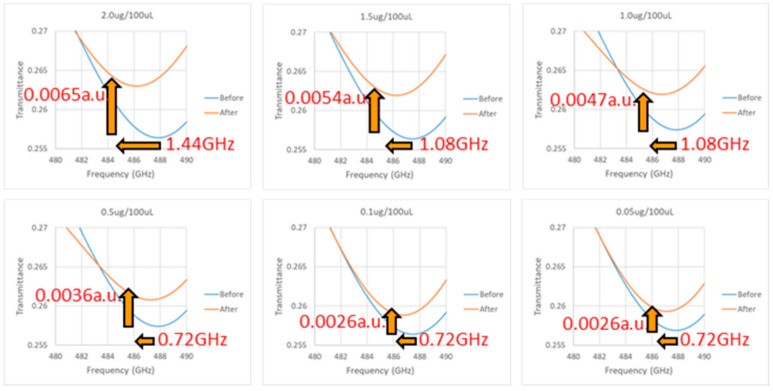
Resonance curve of troponin antigen experiment by VDI system.

**Figure 10 micromachines-13-02257-f010:**
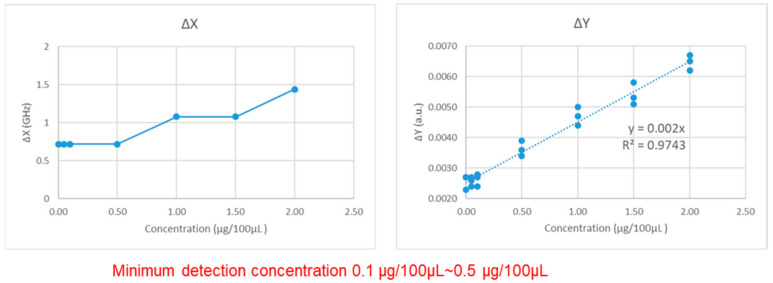
Relationship between the antigen concentration and ΔX.ΔY by VDI system.

**Table 1 micromachines-13-02257-t001:** *Y*-axis resonance position variation in troponin antigen (ΔY) by TeraPulse.

∆Y(a.u.)		Antigen Concentration								
		2.0 mg/100 mL			1.5 mg/100 mL			1.0 mg/100 mL		
After Modification	3 Chips (Each)	0.0039	0.0038	0.0040	0.0041	0.0040	0.0040	0.0040	0.0038	0.0038
	Average	0.0039			0.0040			0.0039		
After Antibody	3 Chips (Each)	0.0030	0.0029	0.0031	0.0030	0.0030	0.0030	0.0031	0.0028	0.0028
	Average	0.0030			0.0030			0.0029		
After Antigen (Corresponding)	3 Chips (Each)	0.0023	0.0022	0.0023	0.0018	0.0017	0.0016	0.0012	0.0013	0.0011
	Average	0.0023			0.0017			0.0012		
**∆Y(a.u.)**		**Antigen Concentration**								
		**0.5 mg/100 mL**			**0.1 mg/100 mL**			**0.05 mg/100 mL**		
After Modification	3 Chips (Each)	0.0039	0.0037	0.0040	0.0039	0.0038	0.0039	0.0039	0.0041	0.0040
	Average	0.0039			0.0039			0.0040		
After Antibody	3 Chips (Each)	0.0030	0.0029	0.0031	0.0028	0.0028	0.0031	0.0029	0.0030	0.0031
	Average	0.0030			0.0029			0.0030		
After Antigen (Corresponding)	3 Chips (Each)	0.0007	0.0006	0.0006	0.0002	0.0002	0.0003	0.0000	0.0001	0.0000
	Average	0.0006			0.0002			0.0000		

**Table 2 micromachines-13-02257-t002:** *X-* and *y*-axis resonance shift and variation by VDI system.

		Initial Position		After Binding		Change		Average	
		X Axis (GHz)	Y Axis (a.u.)	X Axis (GHz)	Y Axis (a.u.)	∆X (GHz)	∆Y (a.u.)	∆X (GHz)	∆Y (a.u.)
		Resonant Position	Transmittance	Resonant Position	Transmittance				
Concentration 2.0 μg/100 μL	Chip 1	487.84	0.2561	486.40	0.2628	1.44	0.0067	1.44	0.0065
	Chip 2	487.84	0.2573	486.40	0.2635	1.44	0.0062		
	Chip 3	487.84	0.2564	486.40	0.2629	1.44	0.0065		
Concentration 1.5 μg/100 μL	Chip 1	487.48	0.2565	486.40	0.2618	1.08	0.0053	1.08	0.0054
	Chip 2	487.84	0.2560	486.76	0.2618	1.08	0.0058		
	Chip 3	487.48	0.2567	486.40	0.2618	1.08	0.0051		
Concentration 1.0 μg/100 μL	Chip 1	487.84	0.2575	486.76	0.2619	1.08	0.0044	1.08	0.0047
	Chip 2	487.48	0.2578	486.40	0.2625	1.08	0.0047		
	Chip 3	487.84	0.2571	486.76	0.2621	1.08	0.0050		
Concentration 0.5 μg/100 μL	Chip 1	487.84	0.2573	487.12	0.2609	0.72	0.0036	0.72	0.0036
	Chip 2	487.12	0.2559	486.40	0.2593	0.72	0.0034		
	Chip 3	487.48	0.2563	486.76	0.2602	0.72	0.0039		
Concentration 0.1 μg/100 μL	Chip 1	487.48	0.2563	486.76	0.2587	0.72	0.0024	0.72	0.0026
	Chip 2	487.12	0.2558	486.40	0.2585	0.72	0.0027		
	Chip 3	487.48	0.2572	486.76	0.2600	0.72	0.0028		
Concentration 0.05 μg/100 μL	Chip 1	487.84	0.2573	487.12	0.2597	0.72	0.0024	0.72	0.0026
	Chip 2	487.48	0.2566	486.76	0.2593	0.72	0.0027		
	Chip 3	487.84	0.2558	487.12	0.2584	0.72	0.0026		
Concentration 0.0 μg/100 μL	Chip 1	487.48	0.2561	486.76	0.2588	0.72	0.0027	0.72	0.0026
	Chip 2	487.84	0.2571	487.12	0.2598	0.72	0.0027		
	Chip 3	487.12	0.2569	486.40	0.2592	0.72	0.0023		

## Data Availability

The data presented in this study are available on request from the corresponding author.

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
