# Peer review of "Terahertz Combined with Metamaterial Microfluidic Chip for Troponin Antigen Detection"

_micromachines, 2022, doi:10.3390/mi13122257_

Round 1

Reviewer 1 Report

Please find attached my comments

Reviewer 2 Report

The authors demonstrate terahertz combined metamaterial technology for analytes identification at different concentrations. Overall, this work holds potential for molecule detection and bioanalysis. Some issues should be addressed before further consideration.

1.     Can the authors show the identification specificity?

2.     The authors use Ag-based substrate as the fluidic system, Ag substrate and many other metal noble substrates have already used for plasmonic-based molecular sensing and detection [Nat. Rev. Mater. 3, 228 (2018), suggested to add], can the authors show the advantages of their work compared with plasmonic-based sensing and detection?

Round 2

Reviewer 2 Report

The authors have NOT responded to my previous comments adequately. And have NOT revised the manuscript according to the suggestion!